# Prediction of anemia and estimation of hemoglobin concentration using a smartphone camera

Selim Suner[1,2], James Rayner[1], Ibrahim U. Ozturan[3,¤a‡], Geoffrey Hogan[3,¤b], Caroline P. Meehan[1‡], Alison B. Chambers[4], Janette Baird[1], Gregory D. Jay[1,2,4]*

**1** Department of Emergency Medicine, Brown University, Providence, Rhode Island, United States of America, **2** School of Engineering, Brown University, Providence, Rhode Island, United States of America, **3** Alpert School of Medicine, Brown University, Providence, Rhode Island, United States of America, **4** Department of Medicine, Brown University, Providence, Rhode Island, United States of America

☯ These authors contributed equally to this work.
¤a Current address: Department of Emergency Medicine, Mersin Toros State Hospital, Yenişehir/Mersin, Turkey
¤b Current address: Department of Emergency Medicine, Denver Health Medical Center, Denver, Colorado, United States of America
‡ IUO and CPM also contributed equally to this work.
* gjay@lifespan.org

**Data Availability Statement:** Data are available from the Zenodo database (DOI: 10.5281/zenodo.4926651).

## Abstract

Anemia, defined as a low hemoglobin concentration, has a large impact on the health of the world's population. We describe the use of a ubiquitous device, the smartphone, to predict hemoglobin concentration and screen for anemia. This was a prospective convenience sample study conducted in Emergency Department (ED) patients of an academic teaching hospital. In an algorithm derivation phase, images of both conjunctiva were obtained from 142 patients in Phase 1 using a smartphone. A region of interest targeting the palpebral conjunctiva was selected from each image. Image-based parameters were extracted and used in stepwise regression analyses to develop a prediction model of estimated hemoglobin (HBc). In Phase 2, a validation model was constructed using data from 202 new ED patients. The final model based on all 344 patients was tested for accuracy in anemia and transfusion thresholds. Hemoglobin concentration ranged from 4.7 to 19.6 g/dL (mean 12.5). In Phase 1, there was a significant association between HBc and laboratory-predicted hemoglobin (HBl) slope = 1.07 (CI = 0.98–1.15), p<0.001. Accuracy, sensitivity, and specificity of HBc for predicting anemia was 82.9 [79.3, 86.4], 90.7 [87.0, 94.4], and 73.3 [67.1, 79.5], respectively. In Phase 2, accuracy, sensitivity and specificity decreased to 72.6 [71.4, 73.8], 72.8 [71, 74.6], and 72.5 [70.8, 74.1]. Accuracy for low (<7 g/dL) and high (<9 g/dL) transfusion thresholds was 94.4 [93.7, 95] and 86 [85, 86.9] respectively. Error trended with increasing HBl values (slope 0.27 [0.19, 0.36] and intercept -3.14 [-4.21, -2.07] (p<0.001) such that HBc tended to underestimate hemoglobin in higher ranges and overestimate in lower ranges. Higher quality images had a smaller bias trend than lower quality images. When separated by skin tone results were unaffected. A smartphone can be used in screening for anemia and transfusion thresholds. Improvements in image quality and computational corrections can further enhance estimates of hemoglobin.

**Funding:** J.R. Funded by Brown Emergency Medicine Foundation who played no role in study design, data collection and analysis, decision to publish, or preparation of the manuscript.

**Competing interests:** I have read the journal's policy and the authors of this manuscript have the following competing interests: GD Jay and S. Suner authored US Patent #7,711,403. We confirm that our competing interest (GD Jay and S. Suner authored US Patent #7,711,403) does not alter our adherence to all PLOS ONE policies on sharing data and materials.

## Introduction

Anemia is defined as a low blood hemoglobin concentration. Hemoglobin values which define anemia are determined by the World Health Organization (WHO) and are different for males, females, and children [1]. The classic symptoms of anemia—fatigue, dizziness or lightheaded-ness, headache, shortness of breath, and difficulty concentrating—can lead to major social and economic consequences such as lost wages and medical care [2, 3] contingent on this common co-morbidity. Severe anemia is often a sequela of malnutrition, parasitic infections or underly-ing disease and is a significant risk factor for morbidity and mortality, especially in vulnerable populations such as children, the elderly and the chronically ill [4, 5]. Severe acute anemia may also be caused by blood loss from trauma or other medical conditions such as gastrointestinal hemorrhage. Anemia is widely prevalent, affecting an estimated 5.6% of Americans and more than 25% of the global population [1].

The clinically used gold standard test for diagnosis of anemia is the complete blood count (CBC), which requires trained phlebotomists, laboratory technicians, the use of chemical reagents and dedicated lab equipment [6, 7]. In some rural settings with inadequate access to healthcare, screening for or diagnosing anemia with a CBC may not be economically or logisti-cally feasible. Anemia is disproportionately prevalent in such populations and thus reflects the social determinants of health and the adverse effects of living in resource poor communities [1]. There is an unmet need for inexpensive, accessible, and non-invasive tools capable of screening for and diagnosing anemia.

Cost is perhaps the most formidable obstacle to widespread adaptability of a novel, non-invasive technology. While there have been devices developed which use advanced imaging and spectrophotometric modalities, the fixed cost of implementing novel spectroscopic or reti-noscopic devices for anemia screening is prohibitive for many rural and urban communities. Ideally, a screening method with the greatest likelihood of adoption and success on a global level is one that would use pre-existing technology which is widely prevalent. An estimated 2.7 billion people, or 36% of the world's population, used smartphones in 2019; this number was expected to grow to 2.87 billion by 2020 [8]. Smartphone ownership is growing at a rate faster than that of the global population. Affluent individuals are more likely to own smartphones, but trends suggest that penetrance of smartphones into lower socioeconomic regions is grow-ing steadily worldwide. The Middle East and Africa, which have the lowest estimated rate of smartphone use, are projected to have 13.7% of citizens using smartphones [8]. Use of smart-phone devices for screening for anemia also has the advantage of incorporating this measure-ment into tele-health applications which also utilize these devices.

Non-invasive measurement of hemoglobin concentration is made possible by its biochemi-cal structure. Hemoglobin is a strong chromophore found in mammalian tissue, enabling its evaluation via spectroscopic means. This property has led to research into devices capable of measuring hemoglobin using transcutaneous [9], retinal [10], and mucosal [11, 12] spectros-copy or photography. Each of these methods pose unique challenges. For example, the wide variance in the quantity and abundance of other tissue chromophores such as bilirubin and melanin make the measurement of hemoglobin with transcutaneous spectroscopy less accu-rate. This challenge is amplified when considering the spectroscopic evaluation of ethnically, genetically, and physically diverse populations. The fingernail bed, palmar creases, and con-junctiva are devoid of melanocytes. Hemoglobin measurement using light-based modalities at these sites could be more accurate, given the absence of a "confounding" chromophore. A smartphone application has been developed to measure hemoglobin concentration from the nailbed using digital photography and has an accuracy of ±2.4 g/dL [13]. While promising, these results are unlikely to be generalizable to cyanotic, hypotensive or mildly hypothermic

patients as environmental and physiological conditions significantly alter blood flow to the digits.

The palpebral conjunctiva is a highly vascular mucocutaneous surface with minimal connective tissue between the outer mucous membrane (which is devoid of any chromophores) and blood vessels. In addition to the absence of melanin, there is also no epidermis, dermis or subcutaneous fat which could impede the transmission of light. These layers of tissue are potential confounders that make image analysis of the deeper vascular layers less accurate. The palpebral conjunctival vascular bed is also less affected by environmental and physiological effects of blood flow. Conjunctival pallor can be a sign of severe anemia on physical examination. However, the sensitivity of this finding is highly clinician dependent, with poor inter-observer reliability [14, 15]. Sampling and analysis of digital images of the palpebral conjunctiva can be utilized in estimating anemia [15, 16]. The present study supports development of a non-invasive method of measuring hemoglobin concentration using digital images of the palpebral conjunctiva obtained using a smartphone. This procedure would demand little skill as the only requirements are eversion of the lower eyelid for exposure of the conjunctiva, and image capture minimizing motion, shadow and glare. Subsequent rapid, real-time computation using an image analysis algorithm, which can be processed on a smartphone utilizing an on-device application, accounting for ambient lighting, glare, pigmentation of the surrounding skin, would produce an estimated value for hemoglobin concentration. There have been recent studies in this area using digitized images of the conjunctiva to emulate spectra for super-resolution spectroscopy in the analysis of hemoglobin [17].

Our aim was to develop an algorithm to predict hemoglobin concentration using smartphone captured images of the conjunctiva. We describe the successful development of a novel algorithm that maximizes color resolution and an automated region of interest selector for the conjunctiva, that was used in identifying patients who met WHO defined anemia limits and blood transfusion thresholds post-hoc.

## Materials and methods

### Image collection

This was an observational prospective convenience sample study. All images were collected from patients who were in the ED at Rhode Island Hospital between October 5, 2018 and August 14, 2019 for any chief complaint. The study was conducted using a convenience sample, when trained staff were available to collect images and data. The approximate volume of patients in the ED during the study period was 100,000 visits and approximately 70,000 of these patients did have a blood test for hemoglobin obtained. The study staff decided which patients to enroll based on screening hemoglobin values available in the electronic medical record to ensure a wide range of hemoglobin values across a sufficient number of patients. The Lifespan Miriam Hospital Institutional Review Board approved this study (9/01/2016; 209416). All patients who participated provided written, informed consent. Inclusion criteria were: having a complete blood count (CBC) obtained as part of the patients' care within 4 hours of digital image acquisition, ability to provide informed consent and ability to expose the palpebral conjunctiva of both eyes. Patients with injury or infection of the eye were excluded. Patients could be lying supine or in a sitting position and were asked to remain still and retract their lower eyelid to expose the conjunctiva. Images were obtained under ambient indoor light. For the initial algorithm derivation phase (phase 1) of the study, 32 images were obtained from each patient. Eight images of the conjunctiva with a standard color reference (colorchecker, Passport Photo, x-rite; xritephoto.com) adjacent to each eye with the conjunctiva exposed were recorded in RAW and Joint Photographic Experts Group (JPEG) format

with and without built-in flash. The remaining 24 images were obtained without the color reference and as close to the conjunctiva as could be clearly focused. Care was taken to minimize glare from light sources and minimize movement. Images were obtained using Halide (Chroma Noir LLC, San Francisco, CA) application on an iPhone 7 Plus (Apple Inc, Cupertino, CA). Demographic information including gender, age, vital signs and Massey skin color rating [18] were collected on a data collection form along with hospital laboratory reported lab tests including hemoglobin, total and direct bilirubin (if available) and time of collection. The specific causes of anemia, even if available in the electronic medical record was not included in the database. To reduce variability, all imaging data were collected by a single operator who developed the data collection methods. Image data were downloaded from the smart phone each day and stored on a computer. All data were transferred into MATLAB (Mathworks Inc, Natick, MA) for analysis.

In the validation phase of the study (phase 2) a new cohort of 202 ED patients were imaged while the data collection operator was unblinded to patient hemoglobin values. For these patients, 3 images of each eye, total 6 per patient, were obtained without flash in RAW mode only. All other methodology remained the same as in phase 1 of the study.

Demographic and laboratory data were obtained from the patient's electronic medical record. Massey score (1–9: 1 being light skin and 9 being dark skin) was determined visually by a single observer using a standard skin tone chart and recorded in the database [18].

## Image processing

Images were analyzed using MATLAB. Each RAW image file from the iPhone was initially stored in a Portable Network Graphics (PNG) format file. RAW images provide data directly from the camera sensor without the typical processing and compression that occurs with common formats such as JPEG. In each PNG file there is also a significant amount of metadata regarding time, location and camera settings.

The RAW file was processed using standard techniques to create a MATLAB Red Green Blue (RGB) color format image file. The RAW file enabled custom processing to allow $2^{32}$ levels of color definition. A directory, "RAW Image Directory" (RID), was created to store RAW images gathered for further analysis.

Using techniques described by Rob Sumner [19] a RAW image processing algorithm which maximized the color resolution of images was developed. Each processed image was stored in a custom "JRI" file format which also included the metadata from the original RAW image PNG file.

Each JRI file contains the RAW image stored as a 4032x3024x3 MATLAB matrix. This format is similar to an RGB image where each pixel in the image has a red, green, and blue value to describe its color. In typical RGB images each color channel can have one of 256 values. The custom algorithm used in this analysis generated a higher resolution color definition image. Using RAW data, each color level was encoded with 32 bits giving 4.3 trillion values for each channel and $8x10^{28}$ colors allowing for highly accurate color analysis. Each image from the RID was converted to a JRI file and stored in a Program Data Directory (PDD).

## Database development and image selection

A database was generated, in order to merge and organize information from each RAW image with patient clinical data. Using MATLAB, an application was created which displayed each image for the user to visually inspect and provided a user interface to eliminate **i**mages which are not suitable for processing such as those which partially omit the conjunctiva, have poor lighting or are completely out of focus. For all valid images, the user was prompted to select a

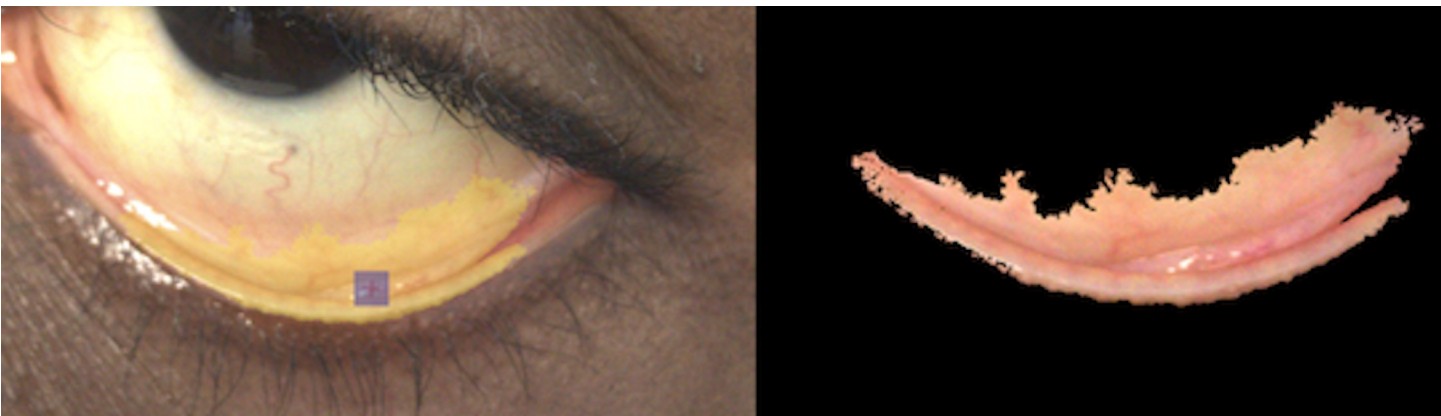

**Fig 1. Image capture and region of interest selection.** (A) Participant's left eye in the MATLAB application for selection of conjunctiva region to be used in analysis. The gray square outlines the selected pixel representing best conjunctiva color. (B) Region of interest (ROI) of the patient's palpebral conjunctiva that was selected by the ROI algorithm.

point within the palpebral conjunctiva representative of overall conjunctiva color, with a mouse click. A region of interest (ROI) representing best color was visually selected as a rectangle of standard size around the selected seed point (SP) pixel (Fig 1).

## Region selection and pixel extraction

Images extracted from the ROI were converted to color spaces which provide increased contrast. A software algorithm was developed to select a test area of the conjunctiva, using the SP pixel, by choosing nearby pixels with similar color values in an automated crystallization paradigm using the high contrast boundaries of the white sclera above and the skin below (Fig 1). This selected region was then stored in the PDD database along with patient information including measured hemoglobin. All further analyses of images were conducted using this selected region.

## Parameter extraction

Image-based parameters, 26 in total, were extracted from the ROI (S1 Table). These parameters represent information such as average brightness of the image, average value of red image component, flash activation and entropy of the image. Each parameter set was designated as a row in a table in MATLAB with each column representing the values of a given parameter for each image. The final column in this matrix was populated with the actual laboratory measured hemoglobin (HBl).

Stepwise regression analyses were performed using this matrix to develop a predictive model for estimated hemoglobin from the conjunctiva (HBc) using the Phase 1 derivation data set of 142 patients. Later in a final validation data set of 344 patients, comprised of 142 Phase 1 patients and 202 new Phase 2 patients, we improved the predictive model using k-folds testing iteration on a randomly selected 10% of the total population ten times.

## Image quality

To quantify image quality three separate observers (two of the authors and one independent observer) rated image quality on a 3-point Likert scale for image domains of focus, extent of conjunctiva exposure and lighting, for each image obtained from 344 patients. Scale dimensions were categorized as "good", "fair" or "bad" for each domain and total points were

calculated for each image from each of the three observers. Gwet's AC1 inter-rater reliability coefficients [20] were calculated for images that were "good", "fair" and "bad".

## Statistical methods

The derivation data set (Phase 1) was used in a generalized linear model for log normal data used to model HBl by the independent variable estimated HBc as well as in flash activation and eye laterality to determine if use of flash lighting improved HBc prediction. A 3-way interaction term was included to allow differences in slope and intercept by each level of flash activation and eye image laterality. Error of HBc to HBl (HBl-Hbc) (normal distribution) was then modeled by flash activation and eye laterality. A two-way interaction was included to allow for differences in slope and intercept. Right and left HBc measures were then averaged for subsequent analysis (RLave). Error of RLave HBc to HBl (HBl-RLave HBc) was then modeled by flash activation. Preliminary interim analysis showed averaging the right and left side HBc images collected without flash best approximated HBl values (S1 Fig).

**Laboratory derived versus estimated hemoglobin.** For the remainder of analysis, only images without use of flash RLave HBc (referred to as HBc for the remainder of analysis) were used in the analysis. Bland-Altman plots (HBl-HBc vs. the mean of HBl and HBc) were used to further assess for bias and precision relative to the gold standard (HBl). Error (HBl-HBc) was modeled by average hemoglobin ((HBl+HBc)/2) to understand underlying trends and with increasing average hemoglobin concentrations. Trends seen (slope and intercept) with the phase 1 derivation data were used to correct predictions for the phase 2 validation data set.

**Clinical usefulness of HBc.** To test clinical usefulness of the HBc measure, HBl and HBc were categorized as anemic (<12.5 g/dL) for women, <13.5 g/dL for men) or not anemic. Agreement between HBl and HBc measures was assessed using 2x2 tables (proc freq) to evaluate the proportion of patients who were categorized as anemic by gold standard HBl versus HBc. HBc was evaluated for significance, sensitivity, specificity, and accuracy. The sensitivity, specificity, and accuracy values were reported as indicators of the usefulness of the HBc in predicting anemia, since p-values alone do not represent the strength of a prediction. This assessment was repeated for blood transfusion hemoglobin cutoffs of 7.0 and 9.0 g/dL. To visualize the tradeoff between sensitivity and specificity with increasing HBc (Receiver Operating Characteristic Curve-ROC and AUC), the proportion of patients who were anemic by HBl were modeled by HBc value (proc logistic).

A general linear model was used to model error of HBc to HBl (HBl-HBc) (normal distribution) by average HB ((HBl+HBc)/2) and image quality. To understand the influence of image quality on the association between HBl and HBc, a two-way interaction was included to allow for differences in slope and intercept by each level of image quality. Bland-Altman plots were also created for the image quality sub-groups. This same analysis was repeated for Massey Score categorized as light skin color (1–3), medium skin color (4–6) and dark skin color (7–9).

All models were analyzed using proc glimmix unless otherwise stated. Nesting for patient and reviewer repeated measures was accounted for by modeling random effects with the residual statement (gee). Methods to correct for the dependence among clustered data were used to adjust for any model misspecification. Familywise error rate (alpha) was maintained at 0.05 using the Holm adjustment for multiple comparisons where appropriate (adjusted p-values are reported). All statistical analyses were performed using SAS version 9.4 (The SAS Institute; Cary, NC).

**Observer agreement.** Gwet's AC1 statistic (first order agreement coefficient) [20] was calculated to assess agreement between observers, adjusting for agreement by chance. This approach to inter-rater reliability addresses a problem often found in calculating observer

agreement using Cohen's Kappa (κ) when skewed or biased (i.e. some raters always rate high or low) distributions of applied ratings results in the inter-rater reliability being substantially lower than the percent agreement among raters [20]. Values of Gwet AC1 inter-rater reliability can be interpreted on the same scale as the κ statistic, <0.40 = poor; 0.40–0.75 = good; > 0.75 = excellent [21].

## Results

During the two phases of enrollment during October 5, 2018 through August 14, 2019, images from 344 unique patients 52% of whom were male were obtained. Hemoglobin concentration (mean 12.5) ranged from 4.7 to 19.6 g/dL. The average age was 53 years with a range of 19–96 years. All patients had pulse oximetry oxygen saturation in the normal range and for those patients who had serum bilirubin ordered (41% of enrolled patients), this value was in the normal range. Mean Massey score (on a 1–9 scale) was 3.6 with a range of 1–9. The distributions of hemoglobin concentration and Massey Score are depicted in Fig 2A and 2B respectively. While the hemoglobin concentration was distributed normally, the Massey score was skewed toward lighter skin color. The selection bias in skin color mirrored the ED population demographics and is consistent with regional US census data for Providence, RI.

For phase 1, 1609 images of the conjunctiva from 142 individual patients were used in the analysis. For phase 2, 1722 images from 337 unique patients were used. All images used in the analyses were RAW images.

One patient withdrew from the study after enrolling and 6 patients had invalid images. 5166 unique conjunctiva area ROI templates were generated by three observers from 1722 images corresponding to 337 unique patient hemoglobin values. Each image was rated by three observers and a total image quality score was compiled for each image. The distribution of image quality scores is shown in S2 Fig.

There was good agreement between the three observers for images from participants with low (image quality score of 6–9; Gwet agreement coefficient 0.6–0.9) and high (image quality score of 16–18; Gwet agreement coefficient 0.6–1.0) quality. Agreement was poor for those which fell in the middle (image quality score 10–15; Gwet agreement coefficient 0.3–0.6).

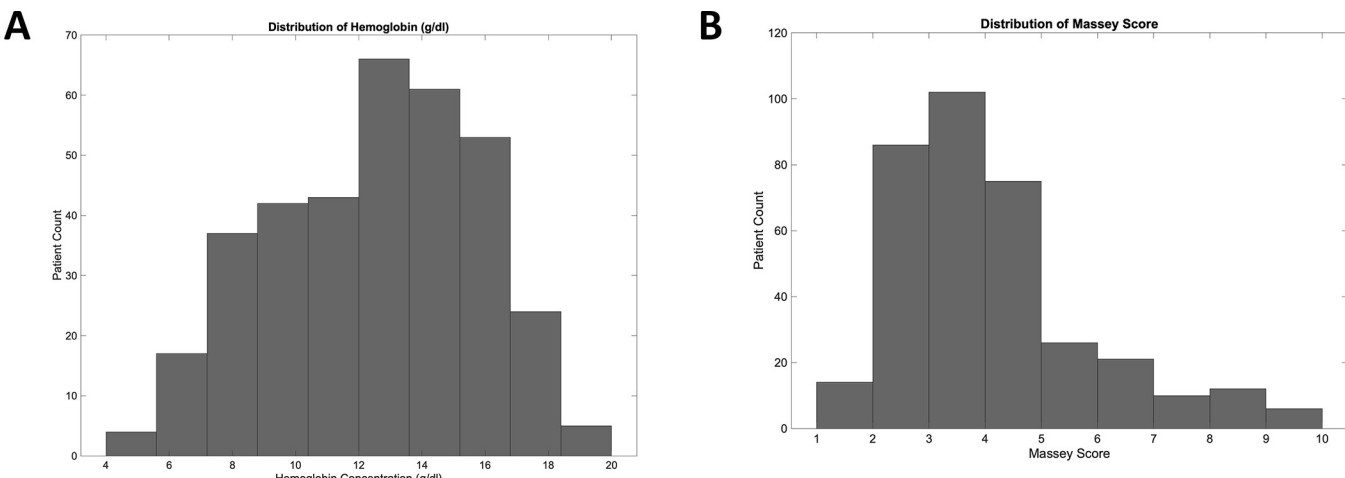

**Fig 2. Distribution of hemoglobin values and skin tones scored across 344 imaged ED patients.** (A) Laboratory assayed hemoglobin concentration was distributed into 2 g/dL bins from 4 g/dL to 20 g/dL. (B) The Massey Score distributed into 9 bins from 1 to 9. Lower Massey scores correspond to lighter skin and higher scores darker skin tone.

## Derivation phase 1 clinical results

The HHR (High Hue Ratio) parameter was the primary parameter in the final matrix that appeared most predictive of HBc following stepwise regression as described above. HBc was significantly associated with HBl (p<0.001) in Phase 1 of our study (Fig 3A). There was a high degree of association between hemoglobin estimated via conjunctiva (HBc) and blood drawn lab values (HBl) slope = 1.07 (CI = 0.98–1.15), p<0.001 (slope of 1 = perfect identity) (Fig 3A). Error (HBl-HBc) for HBc was significantly larger when flash was used (-0.25 [-0.61, 0.1] g/dL) (0.1 [-0.26, 0.47] g/dL, p = 0.0001), prompting elimination of flash use in phase 2 of the study (S1 Fig). Error was not significantly different between the right eye and left eye images (p = 0.2214: -0.15 [-0.52, 0.22] versus 0.00 [-0.36, 0.36], respectively), so the average value from both eyes was used for further analysis.

Accuracy, sensitivity, and specificity of HBc for predicting anemia in males and females was 82.9 [79.3, 86.4], 90.7 [87.0, 94.4], and 73.3 [67.1, 79.5], respectively. Accuracy, sensitivity, specificity, false positive rate, and false negative rate for hemoglobin concentration transfusion thresholds are shown in Table 1. Bland-Altman plot analysis shows a bias of 0.10 and limits of agreement (LOA) of (-4.21, 4.42). Error was found to trend with increasing average HB values (slope 0.27 [0.19, 0.36] p<0.001), where HBc tends to overestimate hemoglobin compared to HBl in the lower range of HB (<11g/dL) (Fig 3B). HBc tends to underestimate hemoglobin compared to HBl in the higher range of average HB (>11 g/dL). This slope and intercept was used to correct for bias in phase 2 of this study; only corrected data is reported in phase 2. A 50% reduction was seen in the bias trend (slope and intercept) for phase 2 with this correction. The tradeoff between sensitivity and specificity with increasing HBc in a receiver operator curve (ROC), for predicting the proportion of patients who were anemic by HBl is shown in Fig 3C.

## Validation phase 2 clinical results

Accuracy, sensitivity, and specificity of HBc for predicting anemia was 72.6 [71.4, 73.8], 72.8 [71.0, 74.6], and 72.5 [70.8, 74.1], respectively. Accuracy, sensitivity, specificity, false positive rate, and false negative rate for hemoglobin concentration transfusion thresholds are shown in Table 1. The Bland-Altman analysis shows a bias of -0.30 and limits of agreement -5.3 to 4.7 g/dL (Fig 4A) in the Phase 2 validation phase of the study. Error was found to trend with increasing

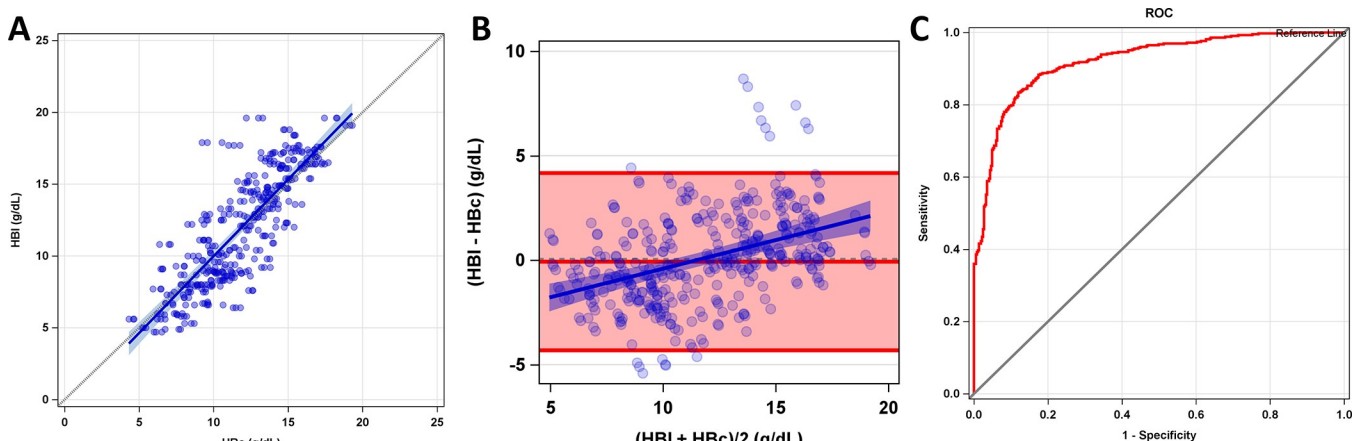

**Fig 3. Phase 1 algorithm derivation study results.** (A) Correlation (identity) plot of HBc and HBl across 142 participants, (B) Bland-Altman plot shows a bias of 0.10 and limits of agreement of -4.21 to 4.42 g/dL, and (C) Receiver Operator Characteristics of HBc using HBl as the gold standard in gender-specific anemia threshold testing. The x-axis depicts 1- Specificity and the y-axis shows the sensitivity. The red line represents the ROC and the black line is the no-discrimination line. Area Under the Curve (AUC) = 92.1.

**Table 1.  Clinical usefulness of conjunctiva determined HB determined from derivation (Phase 1) and validation (Phase 2) data sets.**

| Predicted Outcome | HBc predicting HBl | Phase 1 | Phase 2 |
|---|---|---|---|
| | | Estimate [95% CI] | Estimate [95% CI] |
| **Anemia** | Accuracy | 82.9 [79.3, 86.4] | 72.6 [71.4, 73.8] |
| <**12.5 g/dL Women** | Sensitivity | 90.7 [87, 94.4] | 72.8 [71, 74.6] |
| <**13.5 g/dL Men** | Specificity | 73.3 [67.1, 79.5] | 72.5 [70.8, 74.1] |
| | False Positive Rate | 26.7 [20.5, 32.9] | 27.6 [25.9, 29.2] |
| | False Negative Rate | 9.3 [5.6, 13] | 27.2 [25.4, 29] |
| **Transfusion Low** | Accuracy | 91.1 [88.3, 93.9] | 94.4 [93.7, 95] |
| <**7 g/dL** | Sensitivity | 40 [25.7, 54.3] | 9.3 [5.9, 12.7] |
| | Specificity | 97.5 [95.9, 99.1] | 99.2 [99, 99.5] |
| | False Positive Rate | 2.5 [0.9, 4.1] | 0.8 [0.6, 1.1] |
| | False Negative Rate | 60 [45.7, 74.3] | 90.7 [87.3, 94.1] |
| **Transfusion high** | Accuracy | 81.4 [77.6, 85.2] | 86 [85, 86.9] |
| <**9 g/dL** | Sensitivity | 51.2 [42.4, 60.1] | 39.6 [36.4, 42.7] |
| | Specificity | 94.6 [92, 97.3] | 96.1 [95.5, 96.7] |
| | False Positive Rate | 5.4 [2.7, 8] | 3.9 [3.3, 4.5] |
| | False Negative Rate | 48.8 [40, 57.6] | 60.4 [57.3, 63.6] |

Accuracy, sensitivity, specificity, false positive rate, and false negative rate for anemia as defined by the World Health Organization (WHO) and for transfusion thresholds are shown in this table. Hemoglobin values are in g/dL. Accuracy, sensitivity, specificity, false positive rate, and false negative rate are shown as predicted values [95% confidence intervals].

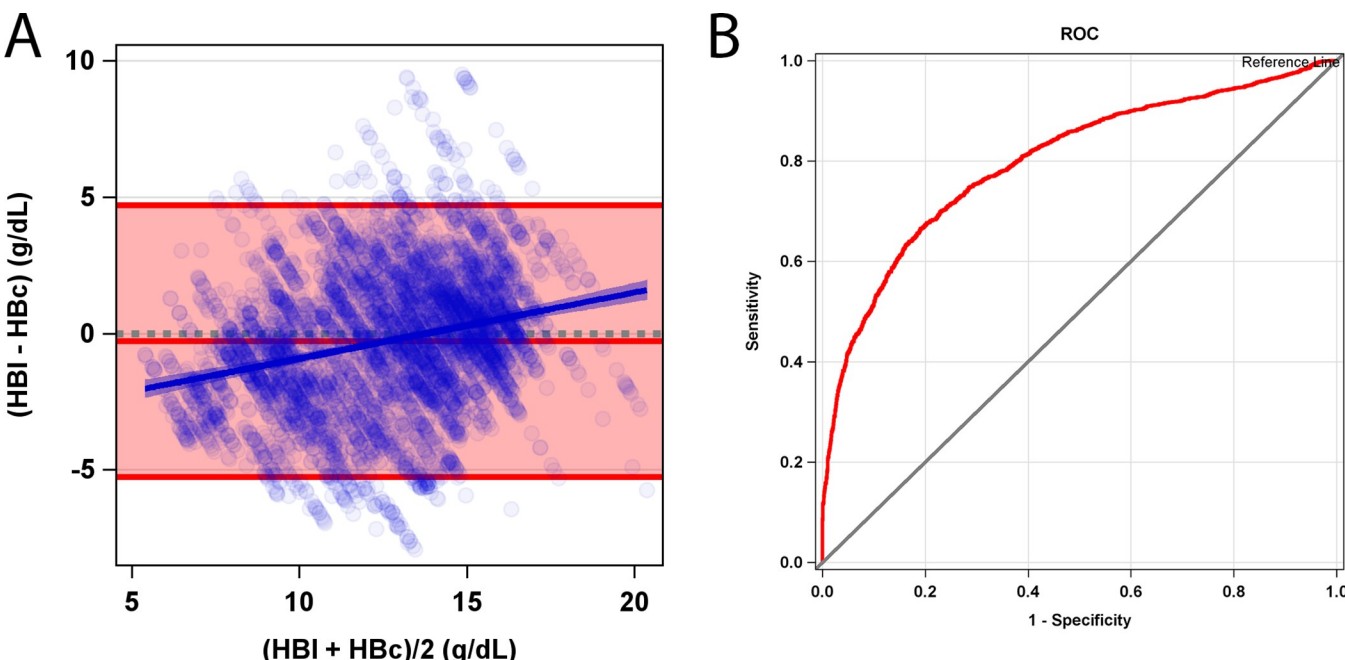

**Fig 4. Accuracy results and clinical utility of Phase 2 algorithm validation study.** (A) Bland-Altman plot for HBc compared to HBl for all 344 participants in the Phase 2 validation data. The average hemoglobin concentration (HBl+HBc/2) in g/dL on the x-axis is plotted against hemoglobin concentration error (HBl-HBc) in g/dL on the y-axis. The pink shaded area represents limits of agreement (Bias = -0.3, upper LOA = 4.7, Lower LOA = -5.3). Model fit for error by increasing average hemoglobin concentration is represented by the solid blue line. Blue shaded region represents slope 95% confidence intervals for error fit. Grey dotted line represents 0 error. (B) Receiver Operator Characteristics of HBc using HBl as the gold standard in gender-specific anemia threshold testing. The x-axis depicts 1- Specificity and the y-axis shows the sensitivity. The red line represents the ROC and the black line is the no-discrimination line. Area Under the Curve (AUC) = 0.80.

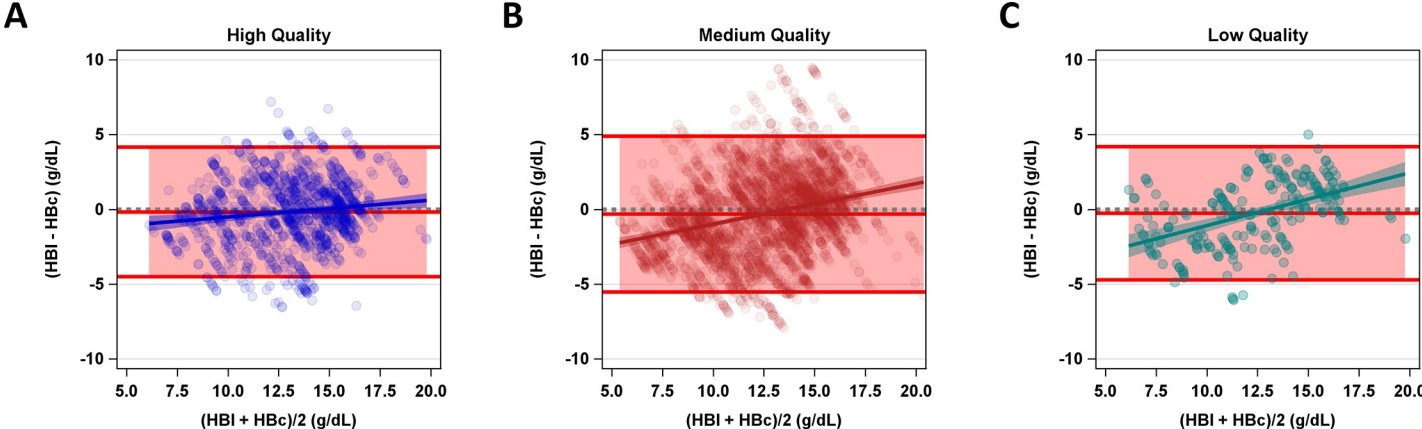

**Fig 5. Bland-Altman plots for HBc compared to HBl separated by image quality.** The average hemoglobin concentration (HBl+HBc/2) in g/dL on the x-axis is plotted against hemoglobin concentration error (HBl-HBc) in g/dL on the y-axis. The pink shaded area represents limits of agreement. Model fit for error by increasing average hemoglobin concentration is represented by the solid lines. Shaded region represents slope 95% confidence intervals. Gray dotted line represents 0 error. The different colors represent data from different image quality ranges. Blue (a) represents high quality image data, red (b), medium quality and green (c), low quality. Colored lines show the respective slope and 95% confidence interval.

HBl values (slope 0.27 [0.19, 0.36] and intercept -3.14 [-4.21, -2.07] (p<0.001). HBc tends to overestimate hemoglobin compared to HBl in the lower range of HB (<11g/dL). HBc tends to underestimate hemoglobin compared to HBl in the higher range of average HB (>11 g/dL). The tradeoff between sensitivity and specificity with increasing HBc (ROC), for predicting the proportion of patients who were anemic by HBl is shown in Fig 4B.

When image quality was accounted for, error from images with high image quality scores had a smaller bias trend (slope) than medium or low quality images (0.11 [0.05, 0.18], 0.27 [0.22, 0.31], 0.35 [0.25, 0.46], respectively: both comparisons p = 0.0001) and smaller limits of agreement ((-4.5, 4.2), (-5.5, 4.9), and (-4.7, 4.2) g/dL, respectively) (Fig 5).

When data were separated by Massey score grouping, LOA did not appear different between Massey score subgroups. The bias that was observed in the main analysis was also seen in the Massey subgroups. Additionally, the slope of the bias was found to be different between the Massey group 1–3 versus Massey group 4–6 (p = 0.001). No significant difference was detected between Massey group 1–3 versus 7–9 (p = 0.4514) or between Massey group 4–6 versus 7–9 (p = 0.3982) (Fig 6).

## Discussion

Anemia is among the greatest of health care concerns and a common comorbidity in both developed and developing countries. Noninvasive point-of-care testing devices for hemoglobin have been studied by our group [15, 22] and others [12, 17] to avoid venipuncture in determining hemoglobin using standard laboratory methods. Point-of-care testing devices can serve as screens for anemia. Digitized imaging of the nailbeds [13] and conjunctiva [12] have reported acceptable accuracy for point-of-care testing devices which do not necessarily rise to a level of agreement that is +/- 1.0g/dL. A cell phone enabled digital camera designed to acquire images of the conjunctiva would represent an advance over other methods in point-of-care testing since patients are already accustomed to recording "selfie" images that could be used in a medical application. This is an especially attractive opportunity for developing countries which may have sparse, rudimentary and poorly distributed medical systems but are well-interconnected by established telecommunication networks.

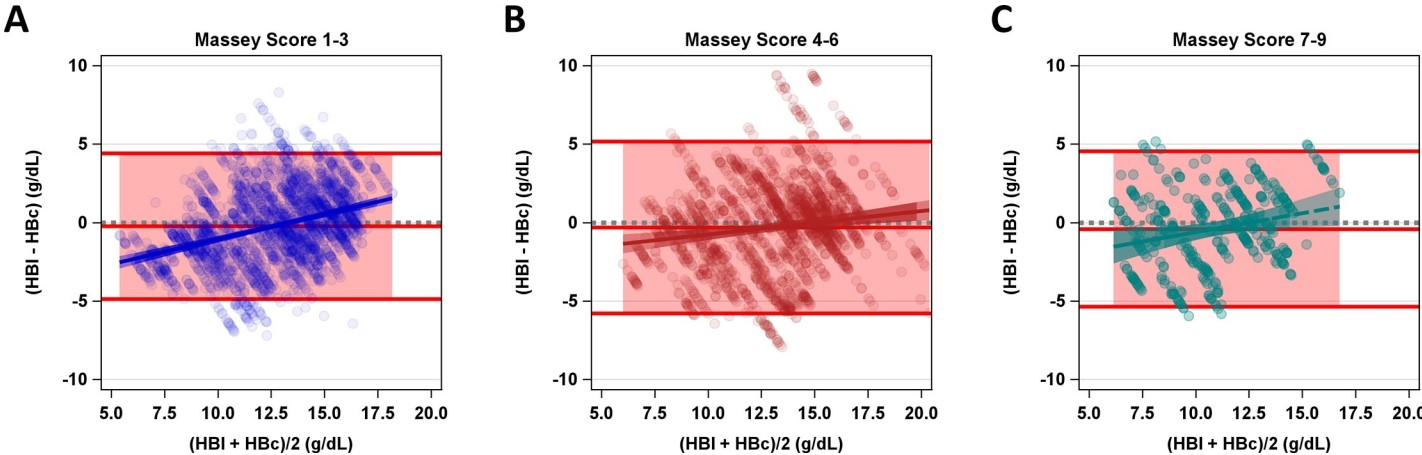

**Fig 6. Bland-Altman plot for HBc compared to HBl separated by Massey score groups.** The average hemoglobin concentration (HBl+HBc/2) in g/dL on the x-axis is plotted against hemoglobin concentration error (HBl-HBc) in g/dL on the y-axis. The pink shaded area represents limits of agreement. Model fit for error by increasing average hemoglobin concentration is represented by the solid colored lines. Shaded region represents slope 95% confidence intervals. Gray dotted line represents 0 error. Each color represents a Massey Score Grouping. Blue (a) representing Massey Scores of 1–3 (light skin), Red (b) 4–6 and Green (c) 7–9 (dark skin) colored lines show the respective slope and 95% confidence interval.

The palpebral conjunctiva was selected for assessment of anemia because of the unique features of this location: A) It is easily accessible for photographic sampling, B)- There are no competing chromophores between the blood vessels and the conjunctival surface to affect color, C) The distance between the tissue surface and blood vessels is very narrow and D) Environmental factors such as temperature do not significantly affect the blood flow to this location.

This study was divided into two phases typical of look-up table algorithmic discovery: 1) Understanding the phenomenon under study; in this case the relationship between known hemoglobin concentration and the spectral content of the conjunctiva using enhanced color discrimination data capture, and 2) Testing both the image capture and the nascent spectral-hemoglobin relationship processing methods against a new population of research participants. The second phase of the study constitutes a real-world data collection set and shows that the level of accuracy was better than or showed equipoise with the hemoglobin estimations provided by pulse co-oximetry [23]. The pulse co-oximeter is a more expensive device which has approval by the U.S. Food and Drug Administration and intended for operating room use for the ongoing intra-operative assessment of hemoglobin. The two staged algorithm development using separate data we described is needed to avoid 'overfitting' of the HBc and HBl association in a derivation dataset alone. Recommendations in development of multivariable prediction models embrace this approach as it reduces bias and offers external validation [24]. It has also been recognized that the number of reported methods described in estimating HBc non-invasively will drive the adoption of technical standards in the future [25]. To our knowledge this is the largest digital conjunctival hemoglobin estimation study to use separate derivation and validation data sets from ED patients with a wide range of hemoglobin values.

In the initial derivation phase 1 of the study we showed that a predictive algorithm can be constructed using elements of a RAW image obtained from a widely available smartphone. We have explored the use of other correlative models but settled on a linear model and the presented variable set. We limited the set to variables we posited as relevant to prevent significant overfitting. Hue saturation value (HSV) also known as hue, saturation, and luminance, and hue, saturation, and intensity are color spaces which mimic human vision closer than RGB

(red, green, blue). Colors perceived to be similar by humans are close together in the color space. By using different color spaces specific image information can be more easily extracted. The HSV is useful as it separated color hue from intensity and saturation allowing appreciation of color changes independent of lighting conditions. High hue ratio, calculated from hue, is a useful calculated variable for hemoglobin prediction which is defined as the number of image pixels with a hue greater that a certain threshold divided by the total number of pixels.

We also showed that adjuncts such as color or white balance references and flash are not needed to construct this predictive model. We showed that there was a good correlation of hemoglobin predicted by this approach to a hospital laboratory confirmed hemoglobin value from blood collected within 4 hours of the acquired image. Applying this algorithm to a larger patient cohort in phase 2 of the study validated the prediction of hemoglobin using Bland-Altman analysis and showed that image quality was a determinant of prediction strength. Poor image quality which is largely operator dependent may explain some loss of accuracy in the Phase 2 validation study. Using an objective Massey skin tone score, we also showed that the prediction was independent of skin color. These results set the stage for the development of an application within a smartphone which can not only acquire the image but also analyze the elements within the image to predict hemoglobin concentration in real-time. The expected and observed independence of HBc from skin tone is an important observation. Recently, pulse oximetry was observed to significantly overestimate SpO2 in African-Americans [26]. This maybe a result of the SpO2 and SaO2 association (look-up) table having insufficient racial diversity. The HBc estimation algorithm we have presented was derived and validated in a diverse ED patient population.

Our group has published extensively on the imaging of the palpebral conjunctiva and its role in estimating hemoglobin concentration noninvasively [10, 15]. The earliest digital cameras available showed the feasibility of this approach. Later we showed that spectroscopy using a grating spectrophotometer would achieve better results [22]. Others have also shown similar results with both cell phone based digital images of the conjunctiva [12] and patient-sourced images of the nail beds using algorithms and images from high resolution digital cameras [13]. Recently, a number of efforts in smaller populations have been reported that attempt to process the conjunctiva RBG image to arrive at a high resolution spectroscopic reconstruction [17], color scale conversion from the RBG image [25] and a logistic regression formula using red, green and blue components [16] in the modeling of hemoglobin concentration. Only one of these studies [17] used a separate derivation and validation model approach. All 3 of these studies used digital images of the conjunctiva and processed RGB images. The accuracy we report using RAW images and 32 bit color level encoding is similar to that of the pulse co-oximeter [27] and other earlier studies using digitized images of the conjunctiva in estimating hemoglobin concentration [28]. Further refinements of this approach using separate algorithm derivation and validation patient cohorts are needed to rigorously determine if this, and related technologies, approximate the 95% LOA within ~ +/- 1 g/dL as some report. Rapid refinements and incremental advancements of in-phone cameras and image processors will likely improve the LOA. An acceptable accuracy for diagnostic point of care devices measuring hemoglobin is in the range of +/- 1.0 g/dL. The reported limits of agreement in our data exceeds this by a significant amount. However, our analyses suggest that the presented data have sufficient sensitivity and specificity to utilize this device as a screening tool for anemia. In its current form the device could be used for screening both inside the hospital and in other settings (home testing, doctor's offices, emergency medical services, etc.) with any abnormal result requiring a confirmatory blood test. We have also shown that image quality has a significant impact on accuracy of predicted results. Further improvements in our imaging technique

and future planned built in corrections based on consistent biases in prediction curves could improve the LOA to an acceptable level for a true diagnostic tool.

The present results also show that RAW image processing coupled with high resolution color encoding performed best in identifying patients who were severely anemic (Table 1) and in need of a transfusion. This may be due to the small bias of our approach that overestimates the hemoglobin level when hemoglobin was low in the first place. Thus, as a screen for clinically significant anemia, the approach we have described appears to identify patients in need of a transfusion at the 9 g/dL threshold. There were too few patients below the 7 g/dL level to fully assess accuracy at that threshold (279 images out of 5166 total images processed). Roughly 90% of these HBc estimates were just above 7 g/dL and reduced sensitivity (Table 1) but are still low and clinically significant enough to trigger a confirmatory venipuncture. The end user would likely respond to low HBc in this manner above or below that threshold. The results of the algorithm at these transfusion threshold levels is not unexpected considering that visual inspection of tissue surfaces for pallor is useful is detecting severe anemia [29].

The present data also shows that no-flash digital photography was adequate to achieve the accuracies we report. We also determined that reviewers of the digital photographs while blinded to the hemoglobin value were able to agree on poor and good appearing data. However, there was disagreement among the reviewers in regard to data that was borderline acceptable. The Gwet AC1 inter-rater reliability analysis in lieu of the kappa statistic confirmed these results which is a common outcome in inter-rater agreement studies [20]. While all images were used in this analysis, future application of this technology could set limits for the use of good images (the user can take multiple images and use the best image) in the analysis increasing accuracy of prediction. Likewise, a refined ROI selection tool and methods to access the camera imager directly, are under development and will also likely improve accuracy. Future studies should also focus on within subject variation of estimated hemoglobin separate from operator variability.

Development of point-of-care digital health technologies has now been accelerated by the need to establish scalable telepresence medical systems in the SARS-CoV-2 pandemic. A cell phone enabled medical application processing the algorithms described above can generate a hemoglobin estimation with acceptable accuracy for point-of-care devices. This technology would fare well with other medical applications created to support remote medical services across a communications network. These networks must be ISO certified, HITech and HIPAA compliant. Telecommunication systems can provide cybersecurity and interoperability in a cellular phone environment and thus enhance smart phone medical applications that utilize embedded sensors. This synergy is separate from traditional telemedicine platforms and offers patients a convenient and socially distant opportunity to access an important laboratory value. Interpretation of the hemoglobin estimation would be the responsibility of the ordering physician; however, the opportunity to measure hemoglobin remotely over a cellular network creates value, particularly for home care-based operations. The number of home care visits has increased significantly over the last several years [30]. Patient-centric approaches in health care will likely gain traction and patient interest beyond the immediate needs created by the global pandemic due to SARS-CoV-2.

Potential limitations include variable image quality which in this case may be due to the fact that the participating patients were retracting their lower eyelids while one of the study co-authors or research assistants recorded conjunctiva images. If both the lid retraction and image acquisition were conducted by the same operator it may have decreased some of the within subject data variability. Any form of standardized lighting was not used in the images that were subsequently analyzed. It is unknown if varying levels of brightness or patient positioning played a role in image quality. Finally, despite the large ED population investigated,

there were small numbers of patients with hemoglobin values at the low and high extremes of measurement.

## Conclusions

We describe a method to estimate hemoglobin concentration using images of the conjunctiva obtained by a smart phone. We demonstrate, using data from patients in the ED, that estimation of hemoglobin concentration by this method can be used as a screening tool for anemia and transfusion thresholds. Furthermore, we show that improvements in image quality and computational corrections can enhance estimates of hemoglobin. There remains a significant step to package image collection, selection and computation into a self-contained application on a smart phone to create a point-of-care device which could be used by clinicians in the hospital or office setting, and by the lay public for screening or to enhance telemedicine encounters. Imaging tools, and further algorithm development, based on *both* model prediction and validation data sets are needed.

## Supporting information

**S1 Table. Table of image-based parameters.** 26 image-based parameters extracted from the ROI. Each image was processed to derive this set of parameters.
(DOCX)

**S1 Fig. Correlation of HBc and HBl between images of the right and left eye in the Phase 1 derivation study.** The graph on the left shows the correlation when flash was used, and the graph on the right shows the correlation without flash. The x-axis depicts HBc in g/dL, the y-axis HBl. The red line corresponds to data from the right eye and the blue line from the left. The shaded areas depict the 95% confidence intervals. The gray dotted line is the identity line (perfect agreement between HBl and HBc).
(TIF)

**S2 Fig. Observer image quality score distribution.** Higher scores correspond to better image quality comprised of the sum of scores for focus, extent of conjunctival exposure, and lighting. Each domain was scored on a 3-point Likert scale and summed across three observers; the maximal score was 18.
(TIF)

## Acknowledgments

We thank Ms. Sandra Spaziano for her help in preparing the manuscript for publication.

## Author Contributions

**Conceptualization:** Selim Suner, Gregory D. Jay.

**Data curation:** Selim Suner, Ibrahim U. Ozturan, Caroline P. Meehan, Alison B. Chambers, Gregory D. Jay.

**Formal analysis:** James Rayner, Alison B. Chambers, Janette Baird, Gregory D. Jay.

**Investigation:** Selim Suner, James Rayner, Ibrahim U. Ozturan, Geoffrey Hogan, Caroline P. Meehan, Gregory D. Jay.

**Methodology:** James Rayner, Alison B. Chambers, Janette Baird.

**Project administration:** Selim Suner, Ibrahim U. Ozturan, Geoffrey Hogan, Caroline P. Meehan, Gregory D. Jay.

**Resources:** Selim Suner.

**Software:** James Rayner.

**Supervision:** Selim Suner.

**Writing – original draft:** Selim Suner, Caroline P. Meehan, Alison B. Chambers, Janette Baird, Gregory D. Jay.

**Writing – review & editing:** Selim Suner, James Rayner, Ibrahim U. Ozturan, Caroline P. Meehan, Alison B. Chambers, Janette Baird, Gregory D. Jay.

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
