## [Decision Letter · Decision Letter 0]

9 Apr 2021

PONE-D-21-07161

Prediction of Anemia and Estimation of Hemoglobin Concentration Using a Smartphone Camera

PLOS ONE

Dear Dr. Jay,

Thank you for submitting your manuscript to PLOS ONE. After careful consideration, we feel that it has merit but does not fully meet PLOS ONE’s publication criteria as it currently stands. Therefore, we invite you to submit a revised version of the manuscript that addresses the points raised during the review process.

Kindly address specifically all comments from reviewer 1.

We look forward to receiving your revised manuscript.

Kind regards,

Benedikt Ley

Academic Editor

PLOS ONE

Additional Editor Comments:

Kindly address specifically the comments of reviewer 1.

Journal Requirements:

[I have read the journal's policy and the authors of this manuscript have the following competing interests: GD Jay and S. Suner authored US Patent #7,711,403.].

Please know it is PLOS ONE policy for corresponding authors to declare, on behalf of all authors, all potential competing interests for the purposes of transparency. PLOS defines a competing interest as anything that interferes with, or could reasonably be perceived as interfering with, the full and objective presentation, peer review, editorial decision-making, or publication of research or non-research articles submitted to one of the journals. Competing interests can be financial or non-financial, professional, or personal. Competing interests can arise in relationship to an organization or another person. Please follow this link to our website for more details on competing interests http://journals.plos.org/plosone/s/competing-interests

5. We note that Figure 1 includes an image of a patient. 

Reviewers' comments:

Reviewer's Responses to Questions

**Comments to the Author**

1. Is the manuscript technically sound, and do the data support the conclusions?

Reviewer #1: Yes

Reviewer #2: Yes

2. Has the statistical analysis been performed appropriately and rigorously? 

Reviewer #1: Yes

Reviewer #2: Yes

3. Have the authors made all data underlying the findings in their manuscript fully available?

Reviewer #1: Yes

Reviewer #2: Yes

4. Is the manuscript presented in an intelligible fashion and written in standard English?

Reviewer #1: Yes

Reviewer #2: Yes

5. Review Comments to the Author

Reviewer #1: Many thanks for the chance to review this fascinating paper. The topic is relevant. The method innovative and the results very important to improve medicine.

I have some issues:

- patient selection: is there a selection bias? recruitment was done between Oct2018 to Aug2019 but only about 344 patients have been included. How many patients would be eligible (admission to ER and blood count? Who did decide which patient should be screened? How do the authors acknowledge heterogeneity and different hb levels.

- anemia may have several underlying causes, iron deficiency,vitamin B12, chronic inflammation, blood loss etc. How dod these types of anemia influence study results?

- acceptable accuracy for point-of-care testing devices is +/- 1.0g/dL

- Fig 3: limits of agreement of -4.21 to 4.42 g/dL is reasonably too high, thus values are noch interchangeable, what does this mean and what would be the benefit of the scan method if values are unacceptable? What were the predefined limits of agreement to consider acceptable LoA? it should be less than 1g/dl, particular if any decision about blood transfusion is planned.

- Fig 4: limits of agreement are even higher, what is the rationale to develop a new measurement technique if values are noch interchangeable or not precise?

What might be the most important medical strength of this new technique?

- decision about anemia yes/no?

- decision about low Hb < 7g/dl

- alternative to invasive blood sampling

Reviewer #2: The paper was written well scientifically good . Lot of efforts has been done. The paper was written well scientifically good . Lot of efforts has been done. The author can compare the conjunctiva output with other anatomical locations

6. PLOS authors have the option to publish the peer review history of their article (what does this mean?). If published, this will include your full peer review and any attached files.

Reviewer #1: No

Reviewer #2: **Yes: **R.MUTHALAGU

---

## [Author Response · Author response to Decision Letter 0]

20 May 2021

Author Comments to the Reviewers:

Reviewer 1

1- We used a convenience sample to recruit patients. This was due to availability of the research staff. We limited data collection to a single staff member to achieve consistency in image collection. The volume of patients coming to our ED during the study period was approximately 100,000 patients. About 70,000 of these patients would have had a CBC test ordered. Our cohort is a small fraction of available patients. The staff member collecting the data decided which patients to screen. The decision involved the hemoglobin value, if available to ensure a wide range of hemoglobin values in our cohort as well as other factors such as if the patient was able to consent, if their care would be affected by enrolling them to the study (if the patient was undergoing specific procedures for instance), etc. The sample was chosen to include a wide range of hemoglobin values including values at the low and high extremes as well those in the normal range to best train our prediction models. When available hemoglobin values were available in the patients’ electronic medical record which the research staff could access. [Edits have been made to the Methods]

2- The underlying cause of anemia was not taken into consideration when recruiting patients, nor was this variable recorded in the research data collection form. Patients likely had a mix of etiologies of their anemia iron deficiency, vitamin B12 deficiency, renal failure, blood loss etc.). We do not believe that the different causes of anemia have a significant effect of our prediction tool. [Edits have been made to the Methods]

3- We agree that an acceptable accuracy for diagnostic point of care devices measuring hemoglobin is in the range of +/- 1.0g/dL. The reported limits of agreement in our data exceeds this by a significant amount. However, our analyses suggest that the presented data have sufficient sensitivity and specificity to utilize this device as a screening tool. In its current form the device could be used for screening both inside the hospital and in other settings (home testing, doctor’s offices, emergency medical services, etc.) with any abnormal result requiring a confirmatory blood test. We have also shown that image quality has a significant impact on accuracy of predicted results. Further improvements in our imaging technique and future planned built in corrections based on consistent biases in prediction curves could bring the limits of agreement down to an acceptable level for a true diagnostic tool. [Edits have been made to the Discussion]

Reviewer 2:

We appreciate the suggestion to compare the conjunctiva output with other anatomical locations.

1- The palpebral conjunctiva was selected for assessment of anemia because of the unique features of this location: A) It is easily accessible for photographic sampling, B) There are no competing chromophores between the blood vessels and the conjunctival surface to affect color, C) The distance between the tissue surface and blood vessels is very narrow and D) Environmental factors such as temperature does not significant affect the blood flow to this location. [Edits have been made to the discussion]

2- The data analysis was conducted using RAW image data files and not JPEG images. In a portion of early data acquisition JPEG images were also acquired in addition to RAW image files, but these data were not used in the analysis.

3- Many different predictive models can be utilized in similar analyses. We have explored the use of other correlative models but settled on a linear model. [Edits have been made to the Discussion]

4- What is sandwich estimation: We have modified the text in the Methods to clarify the methodology: “All models were analyzed using proc glimmix unless otherwise stated. Nesting for patient and reviewer repeated measures was accounted for by modeling random effects with the residual statement (gee). Methods to correct for the dependence among clustered data were used to adjust for any model misspecification. Familywise error rate (alpha) was maintained at 0.05 using the Holm adjustment for multiple comparisons where appropriate (adjusted p-values are reported). All statistical analyses were performed using SAS version 9.4 (The SAS Institute; Cary, NC).”

5- The population size does not significantly affect the sensitivity and specificity in our large cohort of patients. We have intentionally recruited patients with a wide range of hemoglobin values assuring a baseline prevalence of anemia in our cohort.

6- We have intentionally recruited patients with a wide range of hemoglobin values ensuring a baseline prevalence of anemia as well as high hemoglobin values.

7- HSV (also called HSL and HSI) are color spaces which mimic human vision closer than RGB (red, green, blue). Colors perceived to be similar by humans are close together in the color space. By using different color spaces specific image information can be more easily extracted. HSV is useful as it separated color hue from intensity and saturation allowing appreciation of color changes independent of lighting conditions. [Edits have been made to the Results and Discussion sections]

---

## [Decision Letter · Decision Letter 1]

7 Jun 2021

Prediction of Anemia and Estimation of Hemoglobin Concentration Using a Smartphone Camera

PONE-D-21-07161R1

Dear Dr. Jay,

We’re pleased to inform you that your manuscript has been judged scientifically suitable for publication and will be formally accepted for publication once it meets all outstanding technical requirements.

Kind regards,

Benedikt Ley

Academic Editor

PLOS ONE

Additional Editor Comments (optional):

Reviewers' comments:

Reviewer's Responses to Questions

**Comments to the Author**

1. If the authors have adequately addressed your comments raised in a previous round of review and you feel that this manuscript is now acceptable for publication, you may indicate that here to bypass the “Comments to the Author” section, enter your conflict of interest statement in the “Confidential to Editor” section, and submit your "Accept" recommendation.

Reviewer #1: All comments have been addressed

Reviewer #2: All comments have been addressed

2. Is the manuscript technically sound, and do the data support the conclusions?

Reviewer #1: Yes

Reviewer #2: Yes

3. Has the statistical analysis been performed appropriately and rigorously? 

Reviewer #1: I Don't Know

Reviewer #2: Yes

4. Have the authors made all data underlying the findings in their manuscript fully available?

Reviewer #1: Yes

Reviewer #2: Yes

5. Is the manuscript presented in an intelligible fashion and written in standard English?

Reviewer #1: Yes

Reviewer #2: Yes

6. Review Comments to the Author

Reviewer #1: many thanks, no further comment

Reviewer #2: The paper was written well scientifically good . Lot of efforts has been done. only thing so many parameters measured not mentioned in this paper

7. PLOS authors have the option to publish the peer review history of their article (what does this mean?). If published, this will include your full peer review and any attached files.

Reviewer #1: No

Reviewer #2: No

---

## [Editor Report · Acceptance letter]

22 Jun 2021

PONE-D-21-07161R1 

Prediction of anemia and estimation of hemoglobin concentration using a smartphone camera 

Dear Dr. Jay:

I'm pleased to inform you that your manuscript has been deemed suitable for publication in PLOS ONE. Congratulations! Your manuscript is now with our production department. 

Kind regards, 

on behalf of

Dr. Benedikt Ley 

Academic Editor

PLOS ONE